

# A bibliometric analysis of research trends and hotspots in alpine grassland degradation on the Qinghai-Tibet Plateau

Zhe Xu[1], Xian Li[1,2] and Lu Zhang[1]

[1] Research Center for Eco-Environmental Sciences, Chinese Academy of Sciences, Beijing, China
[2] National Plateau Wetland Research Center, College of Wetlands, Southwest Forestry University, Kunming, China

Corresponding author
Lu Zhang, luzhang@rcees.ac.cn

## ABSTRACT

A bibliometric analysis of current research, hotspots, and development trends was used to develop an overall framework of mechanisms of alpine grassland degradation on the Qinghai-Tibet Plateau. This investigation includes data from 1,330 articles on alpine grassland degradation on the Qinghai-Tibet Plateau, acquired from the Chinese Science Citation Database (CSCD) and Web of Science Core Collection (WOS). Research was divided into three themes: spatial scope and management of typical grassland degradation problems, dynamic mechanisms of grassland degradation and effects of ecological engineering, and grassland degradation risk based on remote sensing technology. The results of the analysis showed that the research can be summarized into three aspects: typical grassland degradation identification, dynamic mechanism analysis of grassland degradation, and grassland ecosystem stability strategy. The main findings can summarized, as follows: (1) Ecological analyses using the river source as a typical region defined the formation of "black soil beach" type degraded grasslands in the region, and promoted the ecological environment management and protection of the alpine grassland by discussing the causes of regional ecological environment changes; (2) Dynamic mechanism analyses of climate change and characteristics analyses of grassland vegetation-soil degradation revealed that alpine grassland degradation is the result of multiple main factors; and (3) Risk prediction methods for grassland degradation, methods of grassland management and sustainable countermeasures for agriculture and animal husbandry development, and the development of a comprehensive index of influencing factors on grassland degradation all indicate that selecting the right grassland restoration measures is the key to grassland restoration. Remote sensing monitoring and high-throughput sequencing technology should be used in future research on grassland ecosystems. In addition, multiscale, multidimensional, and multidisciplinary systematic research methods and long-term series data mining could help identify the characteristics and causes of alpine grassland system degradation. These findings can help identify a more effective coordination of landscape, water, lake, field, forest, grass, and sand management for the prevention of alpine grassland degradation.

## INTRODUCTION

The Qinghai-Tibet Plateau is a unique geographical region with the highest altitude in the world (*Zhang et al., 2015a*). The plateau is important to water and soil conservation, as a windbreak and for sand fixation, and in biodiversity protection (*Fu et al., 2021*). Alpine grasslands are an important part of the Qinghai-Tibet Plateau ecosystem that provide essential ecological services for water source protection and climate regulation. They are also important in the global carbon cycle (*Dong et al., 2020*). Alpine grasslands are also the production base and ecological security barrier for animal husbandry in China and are therefore critical in regional economic development, social stability, and cultural inheritance (*Liu, Long & Shang, 2012*). In recent years, owing to the dual effects of global climate change and human activity, alpine grasslands on the Qinghai-Tibet Plateau are experiencing serious degradation (*Hua et al., 2021*; *Xiao et al., 2021*).

Because of continuing grassland degradation on the Qinghai-Tibet Plateau since the 1980s, many studies have been conducted on the causes of this degradation. Climate warming, population increases, grazing pressure, and rodent damage are the key factors leading to grassland degradation (*Harris, 2010*; *Jingsheng et al., 2013*; *Liu et al., 2013*). As a combined result of those factors, coupled with the sensitivity and fragility of grassland ecosystems on the Qinghai-Tibet Plateau (*Hua et al., 2021*), grassland vegetation coverage and biomass in the area have decreased; soil physical, chemical, and biological properties have been damaged; and grassland productivity has gradually declined (*Abril & Bucher, 1999*; *Dlamini et al., 2014*). Some studies indicate that climate change significantly affects alpine grasslands (*Wang et al., 2001*; *Zhang et al., 2017b*). By contrast, according to *Wu et al. (2013)* suggest that the dominant factor affecting grassland health on the Qinghai-Tibet Plateau is not climate change but the intensification of human activities associated with increases in population and grazing intensity. However, grassland degradation likely results from the combined effects of natural factors and human activities. For example, greenhouse gases emitted by overgrazing promote climate warming, which then causes grassland degradation (*Du et al., 2004*). According to *Chen et al. (2014)*, the interactions between climate change and human activities are the dominant factors in grassland degradation. This research suggests the degradation of alpine grasslands is a complex process involving multiple variables and states.

Many studies have examined the degradation of alpine grasslands on the Qinghai-Tibet Plateau, primarily focusing on the composition and structure of degraded grasslands, including species diversity, biomass, ecological vegetation characteristics, community succession, and soil properties (*Liu, Wei-hua & Chen, 2011*; *Wei et al., 2012*). Field quadrangle surveys and spatial analysis are the main methods used to study degradation mechanisms in alpine grasslands, but the results vary depending on the time and data source of the case studies. Summarizing case study results, forming an overall framework of alpine grassland degradation mechanisms, and revealing similarities and differences in study results in different time periods can help identify overall research trends and direct further in-depth research on key ecological processes. A bibliometric analysis was used to objectively and systematically analyze and summarize current research on the hotspots,

dynamics, and development trends in the research on alpine grassland degradation. An overall framework was developed as a reference for the future ecological environmental protection of alpine grasslands on the Qinghai-Tibet Plateau.

## MATERIAL AND METHODS

### Literature acquisition

This investigation includes data collected from 1,330 articles about alpine grassland degradation on the Qinghai-Tibet Plateau published between 1990 and 2022, acquired from the Chinese Science Citation Database (CSCD) and Web of Science Core Collection (WOS). Advanced search methods were used (keywords included: Qinghai-Tibet Plateau OR Tibetan Plateau OR Sanjiangyuan AND Alpine meadow OR Alpine grassland OR Alpine steppe OR Black soil beach AND Degradation OR Deterioration OR Decline). The document type was set to ''Article, Proceedings, Papers, Reviews.''

### Data analysis

Both Citespace 5.7. R2 and Vosviewer 1.6.17 visual analysis software were used to conduct a quantitative analysis of the research on alpine grassland degradation on the Qinghai-Tibet Plateau. Excel 2022 and Origin 2022b software were used for data collation, and for the statistical analyses conducted on the number of published articles, institutions, and authors, and for summarizing keywords. The goal was to analyze international research frontiers and hotspots.

## RESULTS AND DISCUSSION

### General trends in grassland degradation research

A review of the literature found there were 1,330 articles on alpine grassland degradation on the Qinghai-Tibet Plateau published before 2022, including 965 articles found in the Web of Science (WOS) Core Collection and 365 articles found in the China Science Citation Database (CSCD; Fig. 1).

Early research primarily investigated typical phenomena and management of grassland degradation using field studies. In the 1980s, ''black soil beaches'' increased sharply, indicating serious grassland degradation in river headwaters (*Huakun et al., 2003*). In the mid-1990s, permafrost degradation caused by the dry and warming regional climate led to severe vegetation degradation in alpine steppes and meadows, resulting in rapid desertification (*Wang, Shen & Cheng, 2000*). During this period, field investigations conducted in northern Tibet, northwestern Sichuan, southern Qinghai, and Gannan found black soil beaches, indicating degraded grasslands (*Ma & Lang, 1998*). These results indicated that the ecological environments of the grasslands in river source areas were deteriorating, with notable rodent and insect damage, and the degradation of grasslands, primarily indicated by black soil beaches, became increasingly serious (*Yushou et al., 2002*). To prevent further grassland degradation, different management schemes were adopted, including forbidding grazing, controlling poisonous weeds, fertilizing grasslands, and planting artificial or semi-artificial grasslands (*Shang & Long, 2005*).

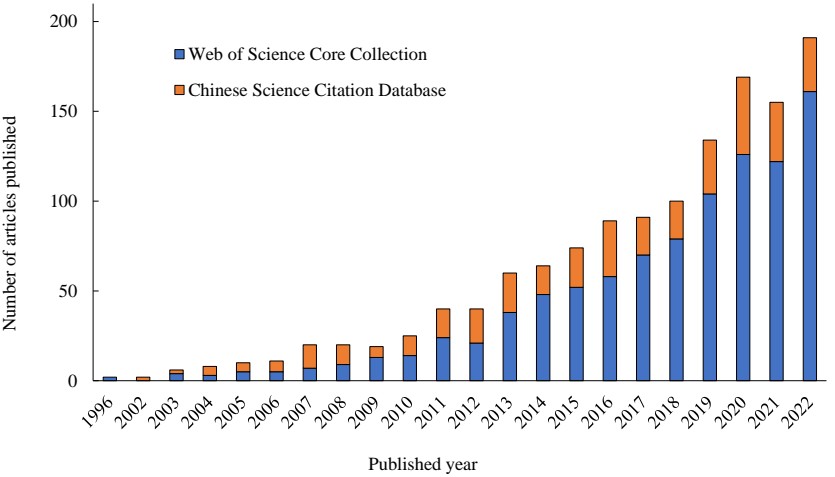

**Figure 1  Trend of published articles.**

Since the implementation of the "Returning Grazing to Grassland" and "Ecological Protection and Construction of the Three-River Source" projects, research on alpine grassland degradation has increased, and the ecological environmental security of grasslands on the Qinghai-Tibet Plateau has attracted great attention. From 2006 to 2015, research focused on: analyzing the causes of grassland degradation (*Cui et al., 2007*), reconstructing ecological environments (*Wu & Du, 2007*), and quantitatively analyzing ecosystem components (*Huakun et al., 2005*; *Jinxia et al., 2003*; *Zhenggang et al., 2004*). To provide a scientific basis for the protection of ecological environments and sustainable use of alpine grasslands, the relationships between plant species diversity, productivity, and soil characteristics were also examined during alpine grassland degradation. Follow-up research focused on determining grassland degradation risk based on remote sensing technology. Responses of grassland ecosystems to climate change were also explored, and sustainable countermeasures in grassland ecological environmental governance and grassland resource protection were implemented based on high-efficiency industrial transformations in both agriculture and animal husbandry. 3S technology (Remote Senescing RS, Geographical information System GIS and Global Positioning System GPS) was used to establish a model for grassland degradation analysis, conduct quantitative research on degraded grasslands in time and space, and predict changes in ecological environments in four-dimensional space (*Li, Perry & Brierley, 2018a*). Metagenomic methods based on high-throughput sequencing technology have also been used to analyze diversity and functional changes in soil bacterial and fungal communities in degraded grasslands (*Zhang et al., 2022*). Research methods in this third stage of research (from 2006-2015) gradually diversified, and the influence of articles greatly increased.

In summary, grassland degradation is a complex process of change, and the clearest changes in the early part of this process are the dramatic increase in black soil beaches and the changes observed in grassland vegetation. Most studies on alpine grassland degradation are time series analyses, outlining changes in grass characteristic indexes to identify the degree

of grassland degradation. Synthesizing the results of these studies, grassland degradation is mainly manifested in the decline of grassland productivity, biodiversity, soil quality, and ecosystem service function. In recent years, with the implementation of the protection and construction project of the ecological security barrier on the Qinghai-Tibetan Plateau, the degradation of alpine grassland has slowed and the ecology of the grasslands have started to recover through the adoption of restoration measures corresponding to the degree of degradation.

### Recognition of alpine grassland degradation in different time periods

Keyword emergence refers to a rapid increase in the frequency of a keyword in a certain period of time, reflecting a research hotspot or new research trend in that period of time. CiteSpace software can extract emergent words from a large number of literature subject words, show the emergence and continuation of relevant research in certain years, characterize the development trend of research hotspots, and reflect the evolution of the research landscape over time (*Xiaonan, Lu & Chunyou, 2014*). Analyzing the emergence of keywords in different time periods can reveal the cutting-edge dynamics of research on alpine grassland degradation on the Tibetan Plateau over time. As seen in Table 1, early research focused on the diagnosis of degradation problems in the typical area of the river source area, and more recent research has focused on the response to degradation of various parts of the alpine grassland ecosystem. The depth and breadth of each aspect of research have increased over time, and new research hotspots have continued to emerge.

In the CSCD database, the keywords "ecological environment" and the research area "river source area" had appeared already prior to the years included in this study, indicating that these were the focus of early research. In 2000, the keyword "ecological environment" appeared, and the research on alpine grassland degradation at this stage focused on evolution and trends in the ecological environment and the causes of regional ecological environment changes (*Huakun et al., 2003*; *Shang & Long, 2005*; *Yang et al., 2005*), which effectively promoted the management and protection of the alpine grassland ecological environment. In the 2003–2007 period, the intensity of "river source area" reached a peak of 5.46 (Table 1). The Three-River Source area provides crucial ecosystem functions and services for Central China and Southeast Asia (*Zhang et al., 2017c*) and is also a typical area for the study of alpine grassland degradation. Because of climate change, nitrogen deposition, and anthropogenic disturbances since the 1970s (*Harris, 2010*; *Dong et al., 2013*; *Dong & Sherman, 2015*) and problems such as grassland degradation, wetland area reduction, glacier retreat, and vegetation coverage declined in the Three-River Source area (*Zhang et al., 2019*), the ecological security of the west is threatened (*Li & Zhang, 2015*). These factors lead to the phenomenon of black soil beaches, which are a sign of ecological environment deterioration in the river park, so "river source area" received high attention in this initial stage of alpine grassland degradation research. During the 2009-2011 period, with an emergent intensity of 4.79 for "climate change," research focused on the influence of climate change on the degradation of grass ecosystem components.

"Soil properties" and "biomass" are the longest lasting research hotspot areas, continuing to be hotspot words even today, as they are a comprehensive reflection of soil physical and

**Table 1** List of burst terms in document analysis.

| Data sources | Burst term | Strange | Begin | End | 1996–2022 |
|---|---|---|---|---|---|
| CSCD | Ecological environment | 2.97 | 2000 | 2006 | |
| | River source area | 5.46 | 2003 | 2007 | |
| | Climate change | 4.79 | 2009 | 2011 | |
| | Soil property | 2.76 | 2009 | 2022 | |
| | Biomass | 2.74 | 2009 | 2022 | |
| WOS | Competition | 2.71 | 2003 | 2007 | |
| | Vegetation | 4.47 | 2014 | 2015 | |
| | Temperature | 3.34 | 2015 | 2017 | |
| | Productivity | 2.9 | 2019 | 2022 | |
| | Pattern | 3.57 | 2019 | 2022 | |

chemical properties and biological attributes that can reveal the most sensitive indicators of soil condition dynamics. *Guo et al. (2019)* analyzed aboveground vegetation and soil nutrients in different degradation and restoration stages of alpine grasslands and found that soil nutrients and their stoichiometry were affected by aboveground biomass. *Wang (2020)* observed an intricate relationship of positive and negative feedback between plants and soils. Soil regulates plant growth, apoplastic production, and vegetation community succession by transporting nutrients and water to plants through the inter-root zone and providing habitat conditions for plant growth and development (*Dahlawi et al., 2018*). Plants exert feedback regulation on soil, and through the input and decomposition of apoplastic matter, soil organic matter content increases, providing nutrients for the growth of soil organisms (*Schittko et al., 2016*).

Emergence words in the WOS database were concentrated from 2003 to 2022 (Table 1), indicating the late start of research on alpine grassland degradation worldwide. Related topics included exploring the effects of degradation factors on grassland plants to reveal the processes and mechanisms of alpine grassland degradation (*Fu, Zhang & Sun, 2019*; *Peng et al., 2020a*; *Zhang et al., 2017a*) and changes in grassland area and vegetation productivity under the effects of climate change and land use patterns (*Gao et al., 2016*; *Sun et al., 2022*; *Wang, Huang & Zhai, 2021*). In the past three decades, primary production of the alpine steppe on the Qinghai-Tibet Plateau has been stable because of changes in the composition of plant functional groups (*Liu et al., 2018*). Natural selection produces differences in plant species diversity in response to environmental changes. Interspecific competition occurs when required environmental resources or energy are insufficient (*Li, Zhu & Xie, 2021*). The persistent emergent keywords in the WOS database focused on grassland productivity and spatial patterns. This result indicated that there is increasing concern about the ecological benefits or ecological services provided by grasslands, which is conducive to building bridges between ecological systems and the socioeconomic system.

In summary, the study of alpine grassland degradation on the Qinghai-Tibetan Plateau has been a gradual evolutionary process. Researchers have sought to identify and analyze the factors and characteristics of degraded grassland from different perspectives and at different times. This study aimed to expand and deepen the response analysis of grassland degradation between various parts of the grassland ecosystem based on an analysis of the keywords, research directions, and publication dates of the relevant articles. This review was more systematic than previous studies.

## Dynamic mechanisms of grassland degradation and effects of ecological engineering

Keywords provide an overall summary of the topic of an article and indicate the core research topics. Thus, research hotspots can be identified through an analysis of keywords (*Yan et al., 2018*). The keywords from the CSCD and WoS databases were merged (Fig. 2), visualized, and analyzed by organizing the keywords in different time periods separately. The first keywords can be summarized as: ecological environment analysis of the region typified by the source of the river. The second time period of keywords can be summarized as: characterization of the vegetation-soil degradation of the grassland, with climate change, grazing activities, and the plateau sage-grouse as the main influencing factors (Fig. 3).

The Qinghai-Tibet Plateau is a unique ecogeographic region characterized by abnormally changing climate and harsh natural conditions that accelerate grassland degradation (*Shao & Cai, 2008*). The effects of climate change on vegetation are complex (*Piao & Fang, 2003*) and include the effects of nitrogen addition rate on species richness and aboveground primary productivity in alpine meadows (*Peng et al., 2022*). Although warming and precipitation, as the main factors of influence, promote the growth of vegetation to a certain extent, precipitation and excessive temperature in summer inhibit the growth of vegetation in alpine areas (*Yanan et al., 2014*), resulting in a decline in forage quality in alpine grasslands (*Dong et al., 2020*). In addition, changes in soil microbial communities in grasslands can cause long-term degradation of the whole ecosystem (*Cui et al., 2007*; *Peng et al., 2020b*). The interaction of warming and soil moisture affect plant growth and forage quality in alpine meadows in the perennial permafrost zone of the Tibetan Plateau, with warming-induced changes seen in community composition, biomass, and forage quality in response to soil moisture availability. Under drought conditions, warming decreased the relative importance of grass and aboveground biomass but increased the importance value of weeds (*Li et al., 2018*). *Duan et al. (2021)* found that the response to climate change varied among different grass types in the Tibetan Plateau region, and that the differences in grass vegetation types resulted in a greater spatial heterogeneity in the response to climate factors among alpine meadows and alpine grasslands. There is also a large spatial heterogeneity in the response of Normalized difference vegetation index (NDVI) to climate factors during the growing season. Open top chambers (OTCs) warming experiments have also been conducted in alpine meadow and alpine grassland habitats in the central Tibetan Plateau, and the results showed that warming led to a significant decrease in the area covered by grasses and forbs and a significant increase in the area covered by legumes,

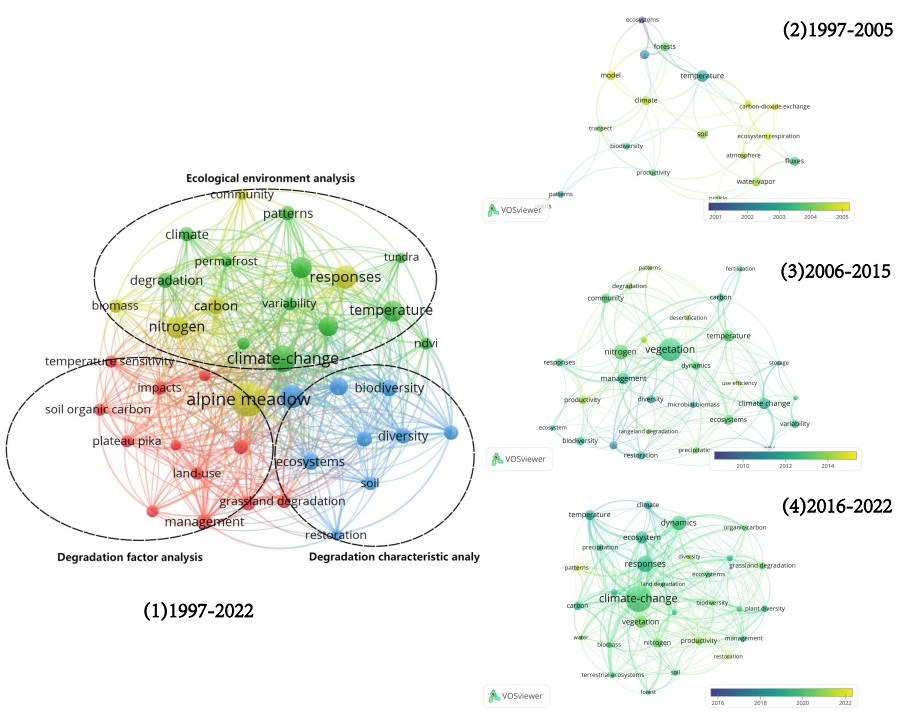

**Figure 2** Cluster graphs of research hotspots in different periods.

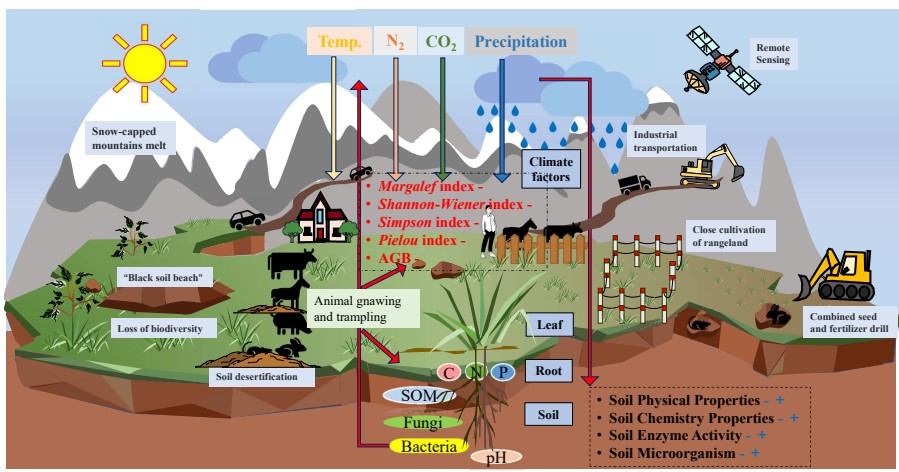

**Figure 3** Overview map of alpine grassland degradation.

leading to a rapid loss of species. Changes in soil moisture were the main cause of these changes in grasses and legumes in alpine grasslands (*Danjiu et al., 2016*). Climate warming

has a positive effect on alpine meadow plant community structure and function, and a negative effect on alpine grassland plant community structure and function.

Livestock grazing affects grassland stability, resilience, and productivity through livestock trampling, foraging, and defecation. Over time, trampling affects a wide range of grassland components and can have lasting effects. Trampling helps keep grasslands healthy but excessive trampling can also lead to the degradation of these grasslands (*Hong et al., 2018*). Cattle and sheep can inhibit growth or kill plants by grazing and trampling, increasing soil hardness, and reducing soil permeability, which is not conducive to plant growth or root development (*Zhao et al., 2013*). Overgrazing seriously threatens the balance of stock–soil–grassland ecosystems and grassland productivity (*Chai et al., 2019*), resulting in the severe degradation of grasslands (*Dong et al., 2020*; *Miehe et al., 2019*), with a loss of biodiversity and natural habitats and accelerated soil erosion (*Filazzola et al., 2020*). High levels of rodent activity can also accelerate grassland degradation. Grazing behavior destroys vegetation, changes community structure, and affects grassland quality. Soil digging behavior changes the distribution of plant biomass and soil structure in grasslands. Such behavior can directly or indirectly affect energy flow and material cycling in alpine grassland ecosystems, leading to the gradual formation of "black soil flats" (*Ma, 1999*; *Smith, Wilson & Hogan, 2019*). Rodents have long been considered the culprits of alpine grassland degradation because their foraging behavior reduces vegetation height and alters plant community structure and biomass, and their digging behavior destroys surface vegetation and increases the risk of soil erosion (*Qin et al., 2016*; *Zhang et al., 2020*).

Overgrazing and climate change are important drivers of alpine grassland degradation in the Tibetan Plateau (*Zhang et al., 2015b*). Climate change inhibits the normal growth of grassland plants, resulting in a decrease in grassland productivity and in the proportion of high-quality pasture grasses, exacerbating the rate of grassland degradation. Continuous grazing causes a decrease in vegetation height and an increase in weed growth, triggering a large increase in plateau zokors and pikas, which further exacerbate grassland degradation by digging and gnawing on the turf (*Li-Zhi et al., 2002*). *Yan, Jay & Brierley (2022)* found that the combined effects of sage-grouse and grazing disturbance reduced aboveground biomass, and that the greater the sage-grouse population, the more intense the competition for food and subsistence space, leading to more biomass loss from these animals nibbling on the soil substrate. In addition, the synergistic effects of grazing activities and pika burrowing affect plant composition, soil properties, and spatial patches in alpine grasslands (*Cao et al., 2021*; *Du & Gao, 2021*; *Wang et al., 2020*). In the Three-River Source region, an appropriate rodent population density is important to the sustainable development of alpine meadows and the protection of grassland resources (*Sun et al., 2011*).

Thus, the degradation of alpine grasslands is the result of interactions between multiple factors, with temperature and precipitation being the primary climatic factors, small herbivorous mammals the main wildlife factor, and livestock overgrazing being the main factor of human activity (*Peng et al., 2018*).

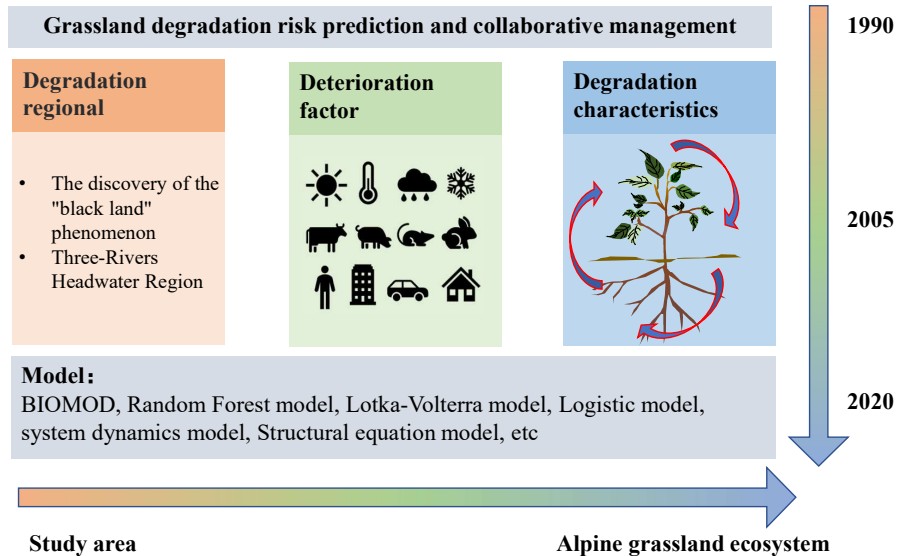

**Figure 4**   Frame diagram for cluster analysis of alpine grassland degradation.

## Strategies for stabilization of grassland ecosystems

The degradation of alpine grassland on the Tibetan Plateau is the result of the joint action of anthropogenic and natural factors, and long-term overgrazing, rodent damage, and climate change have accelerated this degradation (*Huakun et al., 2003*). A cluster analysis framework of alpine grassland degradation (Fig. 4) can be drawn by synthesizing the previous analyses:

In terms of degradation characteristics, the first phase of research focused on the diagnosis of ecological problems in the source area of the river, which was manifested in the serious degradation of alpine grassland with large areas of the surface exposed, changes in soil structure, and a large loss of soil nutrients, stunting vegetation growth and leading to a variety of ecological problems. In the early stage of research on alpine grassland degradation, *Huakun et al. (2003)* analyzed the formation of the black soil beach type of degraded grassland in the source area of the Yangtze River and proposed that the management of degraded grassland in that area should begin by reducing grazing pressure. *Yang et al. (2005)* found that, with degradation leading to black soil beaches, vegetation coverage, aboveground biomass, and the proportion of quality herbage yield decreased significantly with the aggravation of grassland degradation. According to *Shang & Long (2005)* the black soil beach degraded grassland is a unique manifestation of grassland degradation ecological behavior in the alpine grasslands located in the source regions of rivers, and is caused by a combination of factors such as climate warming, glacier retreat, and overgrazing. Revegetation is one of the most effective measures of ecological restoration of black soil beach degraded grasslands, as revegetation can increase the proportion of good forage grasses, improve and restore land productivity, and help control desert spread. Desert spread can be suppressed through desert management and water sand

control, or through other effective ecological control measures (*Quan-Min et al., 2015*; *Xue-Hong, Lei & Gao-Lin, 2010*), which can alleviate the degradation of alpine grassland. The management of black soil beach degraded grasslands is a comprehensive project involving a variety of measures based on degradation degree, and requiring continuous investment and management in order to maintain effectiveness.

Different influencing factors lead to different degrees of grassland degradation. Selecting the right grassland restoration measures is the key to effective grassland restoration. Under mild and moderate degradation, measures such as rodent and insect pest control, fencing and sealing, rotational grazing, fertilizer application, and poisonous weed control should be adopted, while effective grassland restoration strategies, such as artificial intervention to re-establish grassland vegetation or cut through the turf, are needed for moderately and severely degraded grasslands (*Jiang et al., 2020*). Among these strategies, rational fertilization has become an important management measure to maintain nutrient balance, improve plant community structure, and increase grassland productivity (*Zong et al., 2021*). *Li, Zhu & Xie (2021)* found that the addition of biochar and nitrogen improved soil nutrients and microbial communities and helped to restore degraded alpine grassland. However, in some studies (*He et al., 2020*), although restoration measures quickly restored the vegetation coverage and productivity of degraded alpine grasslands, the species diversity of the constructed communities was low because of reseeding with a single species, failure to fully consider the overall ratio of species or functional groups, and unsustainable grassland management measures. Therefore, near-natural restoration methods are advocated in degraded alpine grasslands because these methods fully consider and utilize natural processes and the self-regulation of natural systems to achieve sustainable restoration and stability.

In terms of research methodology, measured data and modeling assessments are still the main tools used in grassland degradation research, with a variety of remote sensing data sources and different interpretation methods being used, combined with ground observations and field surveys to improve the accuracy and reliability of plant phenological period models (*Dong et al., 2020*). With the use of remote satellite tools and mathematical models, the effects of factors such as climate change, grazing activities, and plateau pikas on the degradation of alpine grassland ecosystems and grassland vegetation productivity can be determined more accurately than with previous approaches (*Ganjurjav et al., 2022*; *Gao et al., 2020*; *Kong et al., 2019*; *Wang, 2016*). *Zhao et al. (2022)* used the vegetation index calculated from Landsat-8 OLI remote sensing data to construct a single factor regression model and a random forest model and then determined the best model for a remote sensing estimation of aboveground biomass. The spatial distribution of grassland biomass in 2019–2021 was obtained by inversion. To better predict the effects of pika distribution in alpine grasslands on grassland degradation, BIOMOD (BIOdiversity MODelling) was combined with remote sensing technology to determine grassland spatial distribution and explore the factors limiting that distribution (*Zhang et al., 2021*). At the eastern margin of the Qinghai-Tibet Plateau, the Biome-BGC carbon cycle model and the SHAW surface model were coupled to predict the interaction between an alpine meadow ecosystem and the atmosphere and to explore the effects of environmental factors on the flux of the

alpine meadow (*Wang et al., 2014*). The net primary productivity of alpine meadows on the Qinghai-Tibet Plateau was also analyzed using the Biome-BGC model (*Qingling & Baolin, 2016*). A spatially explicit simulation model was constructed in an alpine meadow under the influence of grazing using field observations and remote sensing data and then used to explore the long-term dynamics of the alpine meadow community. The results indicated that moderate grazing intensity could maintain normal growth and livestock production and was a sustainable use of grassland and pasture (*Li, Perry & Brierley, 2018b*). Microbial sequencing technology was used to evaluate the degradation degree of alpine grassland and the restoration effects of different restoration measures from the perspective of soil microbial community structure and diversity (*Hu et al., 2022*; *Ma et al., 2022*; *Wang et al., 2021*; *Yu et al., 2022*). It is evident from the existing research that understanding and predicting the response of soil microbes and vegetation to environmental variables and the impact of the ecosystem services they provide is both a great challenge and a major research opportunity.

The degradation and succession process of alpine grasslands on the Tibetan Plateau is closely related to the fragile ecological environment of this area, as well as climate change and human activity. These grassland ecosystems have produced different responses from various parts of the ecosystems when faced with external stresses. Future research should explore the various vegetation and soil change characteristics manifested in the grassland degradation process and study and formulate the criteria for using remote sensing to evaluate grassland degradation at the regional scale based on these vegetation and soil characteristics. Although remote sensing can quickly and objectively assess the current status of grass growth, it is currently impossible to exclude information such as increased vegetation cover and grass productivity caused by poisonous weeds. Therefore, applying grass productivity or vegetation cover indicators to the quantitative characterization of vegetation communities is necessary to obtain accurate information on grassland degradation. Future research should also focus on: integrating the influencing factors of grassland degradation into a comprehensive index, using spatial analysis techniques and ecological risk assessment to measure the potential risk of alpine grassland degradation caused by human activities and extreme climate changes and evaluate the main sources of risk, exploring and predicting the formation mechanism of ecological risks and mitigating the losses brought about by those sources of risk. Research scales in this research area continue to broaden from local to regional and global, with expanding spatial scales of research, and lengthened time scales.

## CONCLUSIONS

This article introduces a general framework of the hotspots and development trends in research on alpine grassland degradation on the Qinghai-Tibet Plateau. Research on alpine grassland degradation was divided into three themes: revealing typical degraded grasslands, analyzing the dynamic mechanisms of degraded grasslands, and studying degradation risk based on remote sensing monitoring and ecological models. In earlier research, a sharp increase in the area of "black soil beaches" was found in river source areas, revealing

the characteristics of grassland degradation. In recent years, studies have focused on the responses of each part of the alpine grassland ecosystems, identifying climate change as the main factor of influence on the decrease in productivity of grassland vegetation communities. Community structure degradation and declines in both soil water-holding capacity and nutrient levels in grasslands have also been investigated.

Although major research hotspots and their evolution were identified, more in-depth information is still needed about each hotspot. Therefore, the following future research priorities are recommended: (1) characterize the relationships between soil microorganisms and vegetation with the help of remote sensing monitoring technology and high-throughput sequencing technology and develop remote sensing identification technology for grass species to improve the precision of identifying affected alpine grassland ecosystem processes; (2) examine changes in alpine grassland ecosystems based on long-term data mining to better understand the characteristics and causes of alpine grassland degradation and establish bridges between grassland ecosystems and the socioeconomic system; and (3) use a system perspective to understand the key mechanisms of multi-type process coupling, multi-factor regulation, and multi-scale effects on the relationships between the socioeconomic system and grassland ecosystems. This summary of current research progress and hotspots provide a reference for future research on alpine grassland degradation on the Qinghai-Tibet Plateau.

### Funding
This work was financially supported by the National Natural Science Foundation of China (42171285), the Second Scientific Expedition to the Qinghai-Tibet Plateau (2019qzkk0308), and the Pilot Project of the Chinese Academy of Sciences (XDA20020402). The funders had no role in study design, data collection and analysis, decision to publish, or preparation of the manuscript.

### Grant Disclosures
The following grant information was disclosed by the authors:
The National Natural Science Foundation of China: 42171285.
The Second Scientific Expedition to the Qinghai-Tibet Plateau: 2019qzkk0308.
The Pilot Project of the Chinese Academy of Sciences: XDA20020402.

### Competing Interests
The authors declare there are no competing interests.

### Author Contributions
- Zhe Xu performed the experiments, analyzed the data, prepared figures and/or tables, authored or reviewed drafts of the article, and approved the final draft.
- Xian Li analyzed the data, prepared figures and/or tables, and approved the final draft.
- Lu Zhang conceived and designed the experiments, authored or reviewed drafts of the article, and approved the final draft.

## Data Availability

The raw bibliometric data are available in the Supplemental Files.

## Supplemental Information

Supplemental information for this article can be found online at http://dx.doi.org/10.7717/peerj.16210#supplemental-information.

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
