# Peer review of "A bibliometric analysis of research trends and hotspots in alpine grassland degradation on the Qinghai-Tibet Plateau"

_PeerJ, doi:10.7717/peerj.16210_

## Round 0.1 · original submission · Major Revisions

Dear Authors

Two anonymous reviewers - and I agree with them - highlight a number of critical issues that should be addressed. In particular, reviewer 2 highlights important critical issues that must be adequately addressed in the new version of the manuscript.

Reviewer 1 ·

Basic reporting

The paper analyses the main problems, drivers and influencing mechanisms of alpine grasslands on the Tibetan Plateau and reviews the relevant ecological stabilization measures. It is of great significance for the sustainable management of alpine grasslands in this ecologically sensitive area of the Tibetan Plateau, but there are also the following problems:
1.There are a large number of qualitative descriptions in the paper, and although the review is difficult, the quantitative analysis of the results is crucial to grasp the dynamics of research in the region, rather than simply being 'most' (line 206)
2. Line 132 of the thesis, 'Identification and comprehensive management of typical grassland degradation' is more about the identification of alpine grassland degradation on the Tibetan Plateau and less about comprehensive management. It overlaps with the following paragraph "Strategies for stabilization of grassland ecosystems", and it is suggested that the title of this chapter be revised and the content modified accordingly.
3. Also in the identification of alpine grasslands on the Tibetan Plateau, there is too much text describing the factors influencing alpine wetlands in the region, whereas the focus of this chapter should be on the different stages of development of alpine wetland identification.
4. Line 183, it is important to identify the mechanisms of land degradation in alpine grasslands on the Tibetan Plateau, and although Figure 3 clearly shows the mechanisms of land degradation, Unfortunately, I do not see the interrelationships and potential impacts between the different influencing mechanisms in the text.
5. There is a logical overlap between the different paragraphs in the text, line 232 states that "Degradation of alpine grasslands is primarily manifested in three aspects. In fact, these elements should be found in the section on land degradation identification
Line
6.Line 246 "Strategies for stabilization of grassland ecosystems" is a section that should be presented in the context of the above-mentioned identification of features, degradation mechanisms, and the main implications of this review. Strategies" should be presented in the context of the above-mentioned character identification, degradation mechanisms, and the main implications of this review, and should not be simply listed together, with repeated references to such conceptual words as "grazing intensity", "reduced rotational grazing", "appropriate grazing“.

Experimental design

no commen

Validity of the findings

no commen

Reviewer 2 ·

Basic reporting

This manuscript presented us a summary on researches of alpine grassland conservation and management based on numerous published literature. These information is important for us to know the progress of researches on alpine grassland. However, as a research review article, this article did not provide us some new further information based on the summary of pubished literatures. The author listed several articles for many research fields and told us what they did in the research, but, the author did not present us conclusion on these fields. After reading this manuscript, I did not know what is the trend of grassland degradation on Tibet Platerau? and what will happen in the future for the alpine grassland? For the "strategies for stabilization of grassland ecosystems", I did not get the information which strategy is most effective and what we should do to restore the degraded alpine grassland. This manuscript just told us what had been done in previous researches. This not enough for the review article.

Experimental design

Study design is ok.

Validity of the findings

I have talked this issue in section of basic reporting. I think the findings of this manuscript needs to be greatly improved. Finds cannot be just summary of other people's work. New insights should be presented based on previous researches. This is the biggest problem for this manuscript.

Additional comments

Language should be polished by a fluent English speaker.

Annotated reviews are not available for download in order to protect the identity of reviewers who chose to remain anonymous.

---

## Round 0.2 · accepted · Accept

The manuscript is significantly improved compared to the previous version and, therefore, can be accepted for publication.